# Enhancing Visual Domain Robustness in Behaviour Cloning via Saliency-Guided Augmentation

**Zheyu Zhuang**[1], **Ruiyu Wang**[1], **Nils Ingelhag**[1], **Ville Kyrki**[2], **Danica Kragic**[1]

KTH Royal Institute of Technology[1]  Aalto University[2]

zheyuzh@kth.se

**Abstract:** In vision-based behaviour cloning (BC), conventional image augmentations like Random Crop and Colour Jitter often fall short when addressing substantial visual domain shifts, such as variations in shadow, distractors and backgrounds. Superimposition-based augmentations, which blend in-domain and out-of-domain images, have shown promise for improving model generalisation in the computer vision community, but their suitability for BC remains uncertain due to the need to preserve task-critical semantics, spatial-temporal relationships, and agent-target interactions. To address this, we introduce RoboSaGA–a **Sa**liency-**G**uided **A**ugmentation method within the superimposition family, tailored for vision-based BC. RoboSaGA dynamically adjusts augmentation intensity per pixel based on policy-driven saliency, enabling aggressive augmentation in task-trivial areas while preserving task-critical information. Moreover, it integrates seamlessly into existing architectures without requiring structural changes or additional learning objectives. Empirical evaluations in both simulated and real-world settings show that RoboSaGA maintains in-domain performance while significantly enhancing robustness to visual domain shifts, including distractors and background variations, as well as handling lighting and shadow variations. Code available at: https://github.com/Zheyu-Zhuang/RoboSaGA.

**Keywords:** Behaviour Cloning, Visual Generalisation, Data Augmentation

## 1  Introduction

Vision-based behaviour cloning (BC) has made significant strides in transferring behaviour from expert demonstrations to robot skills [1, 2]. These demonstrations involve intricate spatial-temporal interactions between the agent and its environment and often require integration among multiple sensory modalities. However, given the high costs associated with collecting demonstration data, especially in real-world settings [3, 4], datasets often prioritise task-related variability over visual diversity [5, 6]. This prioritisation presents considerable challenges for generalisation under visual domain shifts, such as variations in lighting, shadows, distractors, or backgrounds.

While image-level augmentation techniques such as Random Crop and Colour Jitter preserve the intricacy of BC demonstrations and enhance in-domain performance [7, 8], they fall short in addressing challenges across the broader visual domain (Sec. 4.2). Superimposition-based augmentation techniques, such as selectively removing parts of an image [9, 10] or overlaying images [11], have proven effective in the computer vision community. However, their application in vision-based behaviour cloning remains limited due to the need to preserve task-critical semantics, spatial-temporal relationships, and agent-target interactions. To this end, we introduce RoboSaGA, a **Sa**liency-**G**uided **A**ugmentation designed for vision-based behaviour cloning. It utilises the input saliency back-propagated from visual features as per-pixel blending factors, allowing the preservation of task-critical regions while superimposing the in-domain images onto out-of-domain (OOD) images. Crucially, RoboSaGA does not require additional learnable modules or specific learning objectives, making it readily applicable to multi-view inputs and compatible with various BC policies.

8th Conference on Robot Learning (CoRL 2024), Munich, Germany.

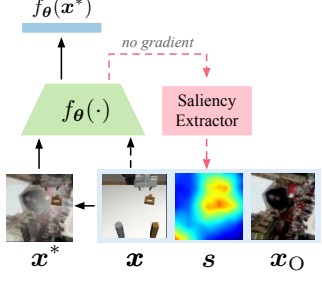

(a) Building Block of RoboSaGA

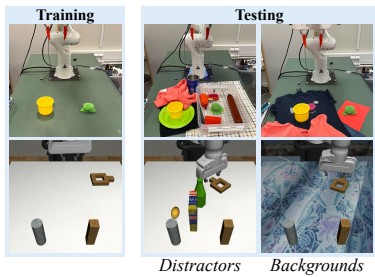

(b) Examples of Visual Domain Shifts

Figure 1: **RoboSaGA's core components and the broadened visual domains.** (a) Saliency map $s$, derived from visual feature output $f_{\boldsymbol{\theta}}(\boldsymbol{x})$, guides the overlaying of in-domain and the OOD image $\boldsymbol{x}_{\mathrm{O}}$ to create the augmented image $\boldsymbol{x}^*$ for training. (b) Examples of training and tested VDS scenarios.

To highlight RoboSaGA's strengths in handling visual domain shifts (VDS), we compare its performance against traditional augmentation methods and existing superimposition-based techniques from the computer vision field within the BC context. Specifically, we evaluate their performance in addressing variations in lighting, shadows, distractors, and backgrounds (with the texture of the operational surface considered part of the background). Performance is measured by the performance gap (Eq. (3)), defined as the difference in mean success rate between the in-domain baseline and the model trained with the augmentation method under visual domain shifts. Experiments span both simulation and real-world settings, focusing on key policy variants: BC-MLP (multi-layer perceptron), BC-RNN (recurrent neural network) [7], and diffusion policy [2]. We summarise our insights:

- Simulated experiments show that both Random Crop and Colour Jitter struggle with distractors and background changes. While Colour Jitter improves performance under light intensity changes, it remains less effective under shadows (gap of 0.33). In contrast, all superimposition-based methods keep gaps below 0.1 under lighting and shadow variations.

- In simulated experiments, the erase-based method effectively handles distractors by replacing task-trivial regions with OOD images but struggles with background variations.

- Random Overlay, applying a constant blending factor for OOD image overlay, reduces the performance gap from Random Crop's 0.60 to 0.24 in simulations and from 0.71 to 0.18 in real-world tests, demonstrating its effectiveness as an augmentation method against VDS.

- RoboSaGA dynamically adjusts augmentation intensity based on saliency maps and further improves performance by reducing the gap to 0.14 in simulations and 0.05 in real-world scenarios, consistently outperforming Random Overlay.

## 2    Related Work

**Vision-based Behaviour Cloning.** A typical vision-based BC policy $\pi_{\boldsymbol{\xi}}\left(\{\boldsymbol{z}_j\}_{j=t-T}^t\right)$, parameterised by $\boldsymbol{\xi}$, derives actions from a temporal sequence of observation embeddings $\boldsymbol{z}$ over a window of length $T$ [7]. At time step $t$, the observation embedding $\boldsymbol{z}_t$ is defined as:

$$\boldsymbol{z}_t := g_{\boldsymbol{\psi}}\left(\{f_{\boldsymbol{\theta}_i}(\boldsymbol{x}_t^{v_i})\}_{i=0}^n, h_{\boldsymbol{\phi}}(\boldsymbol{p}_t)\right),$$

where $\boldsymbol{x}_t^{v_i}$ and $\boldsymbol{p}_t$ represent the visual input from the $i^{\text{th}}$ visual modality or camera view, and the proprioceptive state. The visual encoders $f_{\boldsymbol{\theta}_i}$, parameterised by $\boldsymbol{\theta}_i$, process the visual inputs, while the proprioceptive encoder $h_{\boldsymbol{\phi}}$, parameterised by $\boldsymbol{\phi}$, handles the proprioceptive data. The function $g_{\boldsymbol{\psi}}$, parameterised by $\boldsymbol{\psi}$, integrates these inputs into the latent variable $\boldsymbol{z}_t$. BC policies $\boldsymbol{\pi}_{\boldsymbol{\xi}}(\cdot)$ can be categorised into explicit and implicit types. Explicit policies form the simplest version of BC, directly regressing actions from observation embeddings [12, 13], but they struggle to handle the multi-modal nature of human demonstrations. To address this, Florence et al. [1] reformulate imitation learning as a conditional energy-based modelling problem. More recently, Chi et al. [2] introduce a diffusion-based policy that manages multi-modality by leveraging stochasticity in both the initialisation and sampling processes of diffusion models [14].

**Data Augmentation in Computer Vision.** Data augmentation techniques, traditionally involving randomisations in colour space and image geometry, such as colour jittering, cropping, and rotation, have been expanded with modern superimposition augmentation techniques. Formally, the augmented image $x^*$ is a weighted addition of the in-domain image $x$ and out-of-domain (OOD) images $x_O$, both of the same size $h \times w$:

$$x^* = M \odot x + (1 - M) \odot x_O, \tag{1}$$

where $M \in \mathbb{R}^{h \times w}$ is the blending matrix and $\odot$ denotes element-wise multiplication. $M$ can be uniformly filled with a constant value between zero and one [11], binary with randomly located and sized patches [9, 10], or binary patches guided by input *saliency maps* [15, 16]. $x_O$ can be all-zero matrices [9], random noise [10], or inter-class samples [11, 10]. KeepAugment [15], conceptually closest to RoboSaGA, uses class activation maps to apply a fixed-size rectangular binary mask that covers the most salient areas, preventing the corruption of task-critical regions during augmentation.

**Data Augmentations for Robotic Manipulation.** *With collected datasets*, mild augmentations like Random Crop are essential for improving in-domain generalisability of models [2, 7], though they do not expand the semantic range to cover broader visual domains. Additionally, incorporating language descriptions, Abolghasemi et al. [17] learns language-conditioned masks to exclude task-irrelevant areas. Yu et al. [18] utilise off-the-shelf foundational modules to alter scene attributes, thereby facilitating skill acquisition and enhancing robustness. Superimposition augmentation techniques without task prior have received little attention in BC settings, in contrast to the image-based reinforcement learning (RL) community, where such techniques are more commonly used to enhance policy generalisability [19, 20, 21].

## 3 Saliency-Guided Augmentation for Robotic Behaviour Cloning

RoboSaGA distinguishes itself from other saliency-based superimposition augmentation methods [16, 15] by employing adaptive per-pixel blending factors and extracting saliency directly at the visual encoder level. Unlike KeepAugment [15], which relies on classification logits and fixed-size rectangular crops, RoboSaGA is specifically designed for vision-based manipulation tasks. Its encoder-level saliency extraction ensures modularity and scalability across multiple camera views and policy types, including BC-MLP, BC-RNN [7], and Diffusion Policy [2]. The per-pixel blending approach more effectively handles spatially separated, variable-sized objects and provides greater robustness to visual domain shifts, such as background and lighting changes (Sec. 4.1).

**Saliency Extraction**. RoboSaGA leverages FullGrad [22], which back-propagates the full gradient from the output feature vector back to the full-resolution input, rather than being restricted to gradients from classification logits to specific layers, as in methods like Grad-CAM [23] or Smooth-Grad [24]. FullGrad also incorporates learned biases, in addition to network weights, during saliency computation, resulting in sharper and more accurate saliency maps.

**Saliency Clipping.** RoboSaGA implements a clipping mechanism that caps the normalised saliency scores to a pre-defined threshold $\lambda \in (0, 1)$. By applying a slightly clipped saliency score, RoboSaGA enhances the augmentation of task-critical regions through a subtle overlay, crucial for improving robustness against background changes. See Section 4.1 for details. Formally, given an input image $x^{v_i}$ and its corresponding encoder $f_{\theta_i}(\cdot)$, we define the normalised saliency derived from Fullgrad as $g^{v_i} := \texttt{Fullgrad}(f_{\theta_i}(x^{v_i}))$. The saliency map $s^{v_i}$ is then formulated as:

$$s^{v_i} := \min(g^{v_i}, \lambda) \tag{2}$$

**Data Augmentation**. Within a training batch containing $m$ trajectories (considering a single image as a trajectory of length one), RoboSaGA selectively augments $\alpha$ out of $m$ trajectories, where $\alpha$ serves as a hyper-parameter. This selective augmentation strategy is designed to mitigate potential negative impacts on convergence that can arise from overly aggressive augmentation across the entire batch. Each sample within a trajectory is augmented independently, as no performance gap has been observed when using trajectory-wise augmentation, where all samples in a trajectory share the same augmentation parameters as described in [25].

**Algorithm 1** RoboSaGA Saliency-Guided Augmentation Process

1: $\alpha, \beta$    *# number of trajectories for augmentation and saliency update.*
2: $\lambda \in (0, 1)$    *# saliency clipping factor*
3: **Define class** `RoboSaGA` **with methods:**
4:    `set`(saliency_maps, global_id)    *# update the buffer with new saliency maps*
5:    `get`(global_id)    *# retrieve saliency maps from the buffer*
6:    `get_global_id`(batch_indices)    *# convert batch indices to global IDs*
7:    `get_ood_images`()    *# get random out-of-domain images*
8: **Initialize** `RoboSaGA` instance $\mathbf{S}$ with all-one saliency maps.
9: **for each batch** $\mathbf{M}$ with $m$ trajectories **do**
10:    **for each visual encoder** $v$ in the policy network **do**
11:        **Define** $\mathbf{M}_v$ as the images processed by encoder $v$
12:        *# Update steps*
13:        Sort batch indices $\mathbf{I}$ by saliency buffer update frequency and select bottom $\beta$ as $\mathbf{I}_\beta$
14:        Compute new saliency maps: $\text{new\_maps} = \min\left(\texttt{Fullgrad}\left(v(\mathbf{M}_v[\mathbf{I}_\beta]), \lambda\right)\right)$
15:        Update the buffer $\mathbf{S}.\texttt{set}\left(\text{new\_maps}, \mathbf{S}.\texttt{get\_global\_id}(\mathbf{I}_\beta)\right)$
16:        *# Augmentation steps*
17:        Sample batch indices $\mathbf{I}_\alpha$ uniformly from $\{1, 2, \ldots, n\}$, targeting $\alpha$ trajectories
18:        Retrieve saliency maps for sampled indices: $\mathbf{S}_\alpha = \mathbf{S}.\texttt{get}(\mathbf{S}.\texttt{get\_global\_id}(\mathbf{I}_\alpha))$
19:        Augment images: $\mathbf{M}_v[\mathbf{I}_\alpha] \leftarrow \mathbf{S}_\alpha \odot \mathbf{M}_v[\mathbf{I}_\alpha] + (1 - \mathbf{S}_\alpha) \odot \mathbf{S}.\texttt{get\_ood\_images}()$
20:    **end for**
21:    Optimise the policy with augmented batch $\mathbf{M}^*$
22: **end for**

**Global Saliency Buffer.** While `Fullgrad` produces high-quality saliency maps [22], it incurs a higher computational cost due to the inclusion of both network weights and biases. RoboSaGA mitigates this by employing a global buffer that stores and monitors past saliency maps for each image, significantly reducing computational demands. The buffer is initially filled with ones and updates the saliency for the least recently updated $\beta$ trajectories in each training batch, where $\beta$ is a hyper-parameter. During the augmentation phase, $\alpha$ saliency maps are retrieved from the buffer. To conserve memory, these maps are stored as 8-bit, single-channel images at a reduced resolution. For example, storing maps for 10,000 samples at a resolution of $84 \times 84$ consumes approximately 67 MB of graphic memory. Through experiments, we found this delayed saliency update does not cause observable performance degradation. Analysis of the saliency buffer's performance-computation trade-off can be found in Appendix A. The comprehensive algorithm is detailed in Algorithm 1.

## 4 Experiments

**Manipulation Tasks and Out-of-Domain Settings.** As illustrated in *Fig*.2, in simulations, we use expert human demonstrations from the Robomimic environment [7] to evaluate four tasks of varying complexity and visual diversity: *Lift* (pick), *Square* (pick and insert), *Can* (pick and place), and *Transport* (dual-arm pick, handover, place). Real-world experiments focus on *Toy* (pick and place). Most tasks use a second-person and an eye-in-hand camera, except *Transport* with four cameras, all at $84 \times 84$ resolution. In simulations, lighting is varied by adjusting intensity and colour, shadows are toggled on or off, and distractors are randomised by changing category, quantity, and location. Surface textures and backgrounds are shuffled using materials such as textiles, wood, and ceramics. In real-world settings, distractors are randomly selected from a set of items, and surfaces are altered using textile combinations to change backgrounds. Scenes are randomised per trajectory.

**Tested Policies and Evaluation Protocol.** We evaluated three seminal policies: BC-MLP (Multi-Layer Perceptron), BC-RNN (Recurrent Neural Network), and the state-of-the-art Diffusion Policy [2]. All policies incorporate robot proprioceptive states. BC-RNN and Diffusion Policy also use past observations, while BC-MLP relies on the current state only. Following [7], BC-MLP and BC-RNN employ a Gaussian Mixture Model to capture the multi-modal distribution in human demonstrations. Policies are trained from scratch by default. *In simulation*, the average task success rate is based on

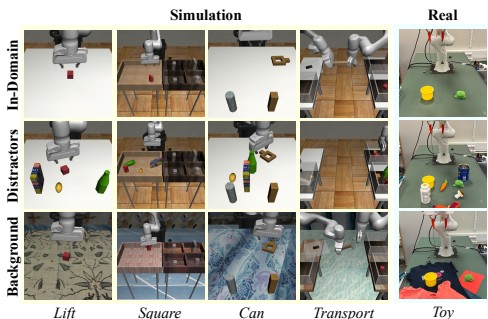

Figure 2: Experiment environment setups.

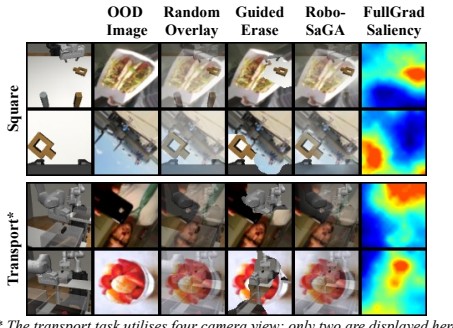

*\* The transport task utilises four camera view; only two are displayed here*

Figure 3: Augmentation with BC-MLP.

the top three checkpoints from 600 training epochs, saved every 50 epochs, and tested over 50 simulation rollouts per checkpoint. *In the real world*, the final checkpoint after 500 epochs is used, with task success rates averaged over 20 trials per data point.

**Implementation Details.** For RoboSaGA, the clipping threshold (Eq. 2) is set to 0.8 across all experiments. The saliency buffer updates 10% of the training batch, and RoboSaGA augments half of the training batch. For OOD images, we use 5000 images from a subset of MSCOCO [26] combined with 1000 synthetic images including plain colours, grids and gradients. The BC-MLP and BC-RNN implementations are identical to the default configurations found in [7]. The Diffusion Policy aligns with the hybrid-CNN in [2]. See Appendix B for experimental setup and implementation details.

**Random Crop as the Default Augmentation**. Random Crop significantly improves both in-domain and out-of-domain generalisation in robotic manipulation [7, 8], making it the baseline augmentation in all experiments. Methods prefixed with "+" denote a combination with Random Crop. All networks use a crop size of 90% of the input, resized back to the original resolution.

**Colour Jitter.** Colour Jitter baselines are obtained by fine-tuning the Random Crop baselines for 50 additional epochs, evaluating checkpoints every 10 epochs, and selecting the top three for comparison. Colour Jitter is implemented with PyTorch's built-in `ColorJitter` function with brightness, contrast, saturation, and hue set to 0.2, as in [8].

**Evaluation Metric**. We assess performance degradation using the performance gap $\Delta\mathrm{P}_{+\mathrm{Aug}}$, defined as the difference between the in-domain mean success rate of the Random Crop baseline and the mean success rate under visual domain shift scenarios with the applied augmentation method. Specifically, the performance gap is expressed as:

$$\Delta\mathrm{P}_{+\mathrm{Aug}} := \mathrm{P}^{\mathrm{In}}_{\mathrm{Baseline}} - \mathrm{P}^{\mathrm{VDS}}_{+\mathrm{Aug}}. \tag{3}$$

This formulation allows for a direct comparison of two augmentation methods by subtraction, effectively cancelling out the baseline performance. A smaller gap indicates greater robustness against the tested visual domain shifts (VDS). Metrics such as mean success rate, performance gap, and standard error are computed using pooled observations across checkpoints, tasks, and policies.

## 4.1 Dissecting RoboSaGA: The Interplay of Overlay and Erase in Behaviour Cloning

RoboSaGA can be conceptualised as a hybrid method that synergistically integrates two distinct augmentation techniques: overlay and erase. Before expanding the scope of our experiments, we undertake an ablation study to dissect the intertwined roles of these techniques within RoboSaGA. The erasing and overlaying variants of RoboSaGA are:

- *Guided Erase:* RoboSaGA binarises saliency maps at a pre-defined threshold for superimposition. This method, designed as a variation of KeepAugment [15], uses variable-sized regions instead of fixed-sized patches and derives saliency from visual features rather than classification logits, enabling more adaptive augmentation.

- *Random Overlay:* The context-aware per-pixel blending matrix from RoboSaGA is replaced by a global constant. This method also appears as a baseline in [20].

|  | BC-MLP | | | |
| --- | --- | --- | --- | --- |
|  | Random Crop | +Guided Erase | +Random Overlay | +Robo-SaGA |
| Background | 0.84 | 0.55 | 0.21 | **0.13** |
| Distractor | 0.52 | **0.06** | 0.31 | 0.06 |
| mean | 0.68 | 0.31 | 0.26 | **0.10** |

Table 1: **Gap (↓) of BC-MLP** under variations in background and distractors.

|  | Average over All Tested Policies | | | | |
| --- | --- | --- | --- | --- | --- |
|  | Crop | +Jitter | +SODA | +Overlay | +SaGA |
| Lighting | 0.39 | 0.10 | 0.05 | 0.02 | **0.01** |
| Lighting & Shadow | 0.56 | 0.33 | 0.08 | 0.09 | **0.05** |
| mean | 0.48 | 0.21 | 0.07 | 0.05 | **0.03** |
| Background | 0.72 | 0.64 | 0.35 | 0.26 | **0.19** |
| Distractor | 0.48 | 0.46 | 0.20 | 0.22 | **0.10** |
| mean | 0.60 | 0.55 | 0.28 | 0.24 | **0.14** |

Table 2: **Gap (↓) across All Policies** (BC-MLP, BC-RNN, Diffusion Policy) under visual domain shifts.

|  | BC-MLP | | | | | BC-RNN | | | | | Diffusion Policy | | | | |
| --- | --- | --- | --- | --- | --- | --- | --- | --- | --- | --- | --- | --- | --- | --- | --- |
|  | Crop | +Jitter | +SODA | +Over. | +SaGA | Crop | +Jitter | +SODA | +Over. | +SaGA | Crop | +Jitter | +SODA | +Over. | +SaGA |
| BG. | 0.75 | 0.65 | 0.38 | 0.27 | **0.21** | 0.80 | 0.74 | 0.40 | 0.27 | **0.21** | 0.60 | 0.52 | 0.29 | 0.24 | **0.16** |
| Dist. | 0.46 | 0.46 | 0.22 | 0.31 | **0.09** | 0.53 | 0.48 | 0.22 | 0.19 | **0.10** | 0.45 | 0.44 | 0.16 | 0.15 | **0.11** |
| mean | 0.60 | 0.56 | 0.30 | 0.29 | **0.15** | 0.66 | 0.61 | 0.31 | 0.23 | **0.15** | 0.52 | 0.48 | 0.23 | 0.19 | **0.13** |

Table 3: **Performance Gap (↓) of BC-MLP, BC-RNN, and Diffusion Policy** with Random Crop, +Random Jitter, +Random Overlay, +SODA, and +RoboSaGA under visual domain shifts.

Both the binarisation threshold for Guided Erase and the blending factor for Random Overlay are set to 0.5. These implementations differ from RoboSaGA only in their approach to superimposition, while the overarching augmentation strategy remains consistent. Specifically, we randomly sample and augment 50% of the trajectories from each training batch for all experiments. *Fig.* 3 illustrates the augmented images from the above methods. We evaluated the efficacy of RoboSaGA and its two variants across three simulation tasks (*Lift*, *Can*, *Square*) with BC-MLP.

As shown in *Tab.*1, Guided Erase fully replaces task-trivial regions, which enhances resilience to distractors, but it leaves task-critical areas untouched, limiting its performance against background variations. On the other hand, Random Overlay uses a universal blending factor, maintaining consistent performance across visual domain shifts but being less effective against distractors due to its milder modification of task-trivial areas. RoboSaGA combines the strengths of both methods, reducing Random Overlay's performance gap under distractors from 0.31 to 0.06, and improving Guided-Erase's performance gap under background variations from 0.55 to 0.13.

## 4.2 The Efficacy of RoboSaGA and Main Observations

In this section, we evaluate each selected augmentation method under visual domain shift scenarios across various behaviour cloning policies in both simulated and real-world settings. *Full tables with task success rates and the standard error of the mean are provided in Appendix C.* This evaluation aims to answer the following questions:

**Q1:** How well do traditional augmentation methods handle visual domain shifts (VDS)?

We evaluated two conventional augmentation methods, Random Crop and Random Crop + Colour Jitter, under four VDS scenarios. *Tab.* 2 presents the performance gap averaged across four tasks and three policies. The results show that lighting significantly degrades baseline performance. While Colour Jitter effectively handles plain lighting changes (intensity and colour), reducing the average gap to 0.1, its improvement is limited under shadows, with a gap of 0.33, compared to less than 0.1 for overlay methods. For distractors and background variation, colour jitter reduces the gap from 0.6 to 0.55, offering only an 8% relative improvement. Given the strong performance of tested superimposition methods in addressing lighting changes, the remainder of this section and the real-world experiments focus on VDS scenarios involving distractor and background variations.

**Q2:** Is representation learning more effective than direct training with augmented images?

In vision-based reinforcement learning, aggressive data augmentation is frequently implemented with representation learning techniques for improved policy stability [19, 20]. Within this frame-

| | BC-MLP | | | BC-RNN | | | Diffusion Policy | | | All Policies | | |
|---|---|---|---|---|---|---|---|---|---|---|---|---|
| | *Crop* | *+Overlay* | *+SaGA* | *Crop* | *+Overlay* | *+SaGA* | *Crop* | *+Overlay* | *+SaGA* | *Crop* | *+Overlay* | *+SaGA* |
| *Background* | 0.70 | 0.20 | **−0.05** | 0.85 | 0.25 | **0.00** | 1.00 | 0.35 | **0.25** | 0.85 | 0.27 | **0.07** |
| *Distractor* | 0.65 | 0.05 | **−0.10** | 0.70 | 0.20 | **0.15** | 0.35 | **0.05** | 0.05 | 0.57 | 0.10 | **0.03** |
| *mean* | 0.68 | 0.12 | **−0.08** | 0.78 | 0.22 | **0.07** | 0.68 | 0.20 | **0.15** | 0.71 | 0.18 | **0.05** |

Table 4: **Real-world Performance Gap (↓) across Each Policy (and Average)** for Random Crop, Random Overlay, and RoboSaGA on the *Toy* task.

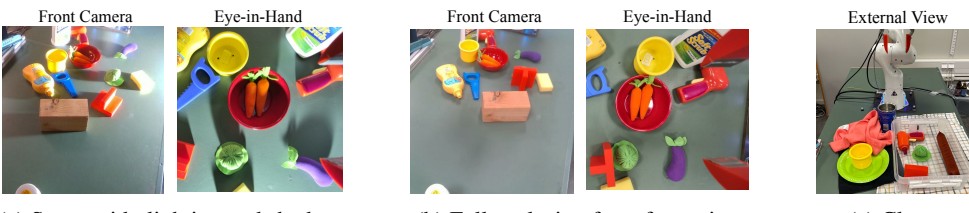

(a) Severe side-lighting and shadow    (b) Full occlusion from front view    (c) Clutter

Figure 4: **Examples of RoboSaGA against Real-World Visual Domain Shifts**, including lighting changes, occlusion, object clutter, and background variations.

work, we aim to determine the necessity and effectiveness of representation learning in behaviour cloning (BC), where action labels are readily available. To this end, we adopted SODA [20] as our baseline, which utilises a student-teacher architecture with a consistency loss to encourage Euclidean similarity between features derived from images augmented by Random Overlay and their original counterparts. We assessed the performance of RoboSaGA and Random Overlay in comparison to SODA using the three policies across four simulation tasks. Experimental results in *Tab.* 2 show that direct training with augmented samples delivers comparable yet marginally improved performance, reducing the performance gap by an average of 0.04 under distractor and background variations, and by 0.02 under lighting and shadows.

**Q3:** Does RoboSaGA's effectiveness against VDS extend to policies beyond BC-MLP?

*Tab.* 2 and *Tab.* 3 tabulate the mean performance gap under visual domain shift (VDS). The results suggest that while performance varies across different policies, all three tested augmentation methods (Random Overlay, RoboSaGA, and SODA) consistently achieve translatable improvements. Notably, the Diffusion Policy exhibits the least performance degradation. These findings are confirmed by real-world experiments, where both Random Overlay and RoboSaGA significantly enhance performance against VDS compared to their Random Crop counterparts (*Tab.* 4).

**Q4:** Does RoboSaGA maintain its performance lead against the computation-free Random Overlay?

Random Overlay is an efficient and effective augmentation method against VDS, reducing the performance gap of Random Crop + Jitter from 0.55 to 0.24 for distractor and background variations, and from 0.21 to 0.06 for lighting and shadow variations. RoboSaGA further improves performance, lowering Random Overlay's gaps to 0.14 and 0.03, with relative improvements of 41.6% and 50.0%, respectively. In real-world experiments (*Tab.* 4), Random Overlay reduces the gap (averaged over three policies) from 0.71 to 0.18 for distractor and background variations. RoboSaGA further reduces it to 0.05, yielding a 72% improvement over Random Overlay.

**Q5:** How effective RoboSaGA is in the real world?

Instead of exploring a wide range of task variants, we focused on a relatively simple 3 degrees-of-freedom pick-and-place task to rigorously evaluate the real-world performance of Random Overlay and RoboSaGA. These methods were tested using BC-MLP, BC-RNN, and Diffusion Policy in challenging real-world visual domain shift scenarios, where distractors appeared as clutter. In addition to the quantitative results reported in **Q4** and *Tab.* 4, we observed highly desirable behaviours from RoboSaGA during testing under severe visual domain shifts across three policies. These behaviours included successful task completion despite aggressive lighting variations, which introduced strong shadows from distractors, and maintaining task execution even with significant occlusions in one view and background changes with added clutter (*Fig. 4*).

| Front Camera  Eye-in-Hand | Front Camera  Eye-in-Hand | Front Camera  Eye-in-Hand |
| :---: | :---: | :---: |
| 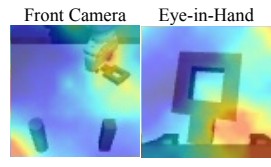 | 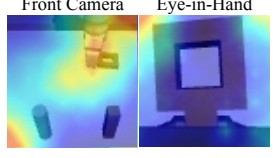 | 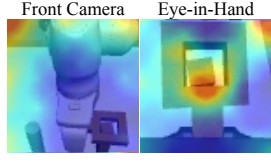 |
| (a) Reaching Nut | (b) Lifting Nut | (c) Inserting Nut |

Figure 5: **Saliency Maps across Two Views** during different stages of task execution (BC-RNN).

**Q6:** How closely do saliency maps correspond to human expectations of task relevance?

Saliency maps do not always align with the robot parts or targets that humans consider critical. As shown in *Fig.* 5a, while reaching for the nut, saliency highlights the general area around the gripper and nut, but the eye-in-hand view focuses only on the handle. When lifting the nut (*Fig.* 5b), the saliency shifts away from the object. During insertion (*Fig.* 5c), the front camera overlooks the target peg and nut, while the eye-in-hand view focuses on the peg and hole. We hypothesise that this salient misalignment with human intuition occurs because vision-based BC integrates multi-sensory inputs (e.g., proprioception, observation history, multi-views) and accounts for task progression. Notably, history-independent BC-MLP produces more human-interpretable saliency maps than its history-dependent counterparts. *Examples are provided in Appendix D.*

## 5   Limitations

**Computation Overhead.** The implementation of a saliency buffer improves efficiency, but the computational demands remain considerable. For example, saliency computation with a `ResNet18` encoder on $84 \times 84$ inputs takes about 1.5 times longer than training. To improve efficiency, approximating saliency at mid-feature layers or exploring alternative saliency extractors could help. For policies with larger encoders, knowledge distillation may offer a solution, where a smaller encoder is trained without augmentation, and saliency is computed in later epochs to create an offline buffer.

**Performance Degradation of BC-MLP and BC-RNN in *Transport*.** As shown in *Tab.* 10, 11, and 12 in Appendix C, while the Diffusion Policy exhibits the expected improvements under the presence of distractors and changes in backgrounds, BC-MLP and BC-RNN demonstrate comparatively lower gains across all tested methods. This discrepancy may arise from the policies' differing abilities to distil task-relevant information from images with high task-related pixel density. As depicted in Fig. 3, the second-person view in the *Transport* task is largely dominated by two manipulators, offering significantly less information about the targets.

**Focused Task Exploration.** Our real-world experiment showcased our approach through a concentrated evaluation of a 3-DoF pick-and-place task, offering valuable insights under specific conditions. Future work could extend these findings to a broader range of tasks.

## 6   Conclusion

In this work, we aimed to enhance the robustness of vision-based behaviour cloning against visual domain shifts, particularly with background changes and object distractors, using superimposition-based image augmentation. Our empirical results from both simulated and real-world experiments demonstrate that overlaying random images onto training data is a cost-effective method that significantly improves performance under visual domain shifts, reducing the performance gap from 0.60 to 0.24 in simulations and from 0.71 to 0.18 in real-world tasks. We introduced RoboSaGA, which uses input saliency maps to adjust augmentation intensity dynamically. This method further reduces the performance gap to 0.14 in simulations and 0.05 in real-world settings, delivering relative improvements of 41.6% and 72%, respectively, compared to Random Overlay. RoboSaGA has revealed the extent of task-relevant semantics hidden within visual features, and the application of it for improving robustness against visual domain shifts. We are excited to explore how these task-specific insights can further advance vision-based behaviour cloning.

**Acknowledgments**

This work has been supported through WASP - Wallenberg AI, Autonomous systems and Software Program. Zheyu would also like to thank Michael Welle for his valuable feedback on the manuscript.

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

# Appendix A: Performance and Computation Trade-off of Saliency Buffer

**Saliency Buffer Implementation Details**

- **Image Global Index:** Each image sample is assigned a global index, which is included as additional data returned by the data loader.

- **Random Crop:** When random cropping (pixel shift) is enabled, cropping randomisers return the cropping parameters, which include the pixel coordinates of the top-left corner and the cropping size. Assuming no additional down-sampling when saving, the buffer maintains the saliency map at its original size before cropping. A saliency map derived from cropped image is padded with zeros in the regions cropped out to match the full image size. Upon retrieval, saliency maps are adjusted to the specified cropping parameters.

- **Pixel Range Conversion:** Each saliency map stored in the buffer is kept as a single-channel 8-bit unsigned integer (UINT8) image to optimise memory use. For augmentation, saliency maps are normalised with pixel values scaled to $[0, 1]$. These values are converted to UINT8 when saved and back to float32 (FLOAT32) within $[0, 1]$ upon retrieval.

- **Saliency Warm-up:** The buffer does not update and returns all-one saliency maps during the first $\gamma$ epochs (default set to 10), allowing the encoders to develop task-specific knowledge before strong data augmentation is applied. This warm-up strategy is also implemented in the RoboSaGA variants: Random Overlay and Guided Erase.

**Performance and Computation Trade-off**

| | Lift | | Can | | Square | |
|---|---|---|---|---|---|---|
| | *Buffer* | *w/o Buffer* | *Buffer* | *w/o Buffer* | *Buffer* | *w/o Buffer* |
| *In-domain* | $0.98_{\pm 0.01}$ | $\mathbf{0.99}_{\pm 0.01}$ | $\mathbf{0.97}_{\pm 0.01}$ | $0.95_{\pm 0.02}$ | $0.68_{\pm 0.04}$ | $\mathbf{0.69}_{\pm 0.04}$ |
| *Background* | $0.89_{\pm 0.03}$ | $\mathbf{0.92}_{\pm 0.02}$ | $\mathbf{0.87}_{\pm 0.03}$ | $0.79_{\pm 0.03}$ | $0.39_{\pm 0.04}$ | $\mathbf{0.43}_{\pm 0.04}$ |
| *Distractor* | $0.95_{\pm 0.02}$ | $\mathbf{0.98}_{\pm 0.01}$ | $\mathbf{0.71}_{\pm 0.04}$ | $0.67_{\pm 0.04}$ | $\mathbf{0.69}_{\pm 0.04}$ | $0.67_{\pm 0.04}$ |
| *mean* | $0.92_{\pm 0.02}$ | $\mathbf{0.95}_{\pm 0.01}$ | $\mathbf{0.79}_{\pm 0.02}$ | $0.73_{\pm 0.03}$ | $0.54_{\pm 0.03}$ | $\mathbf{0.55}_{\pm 0.03}$ |

Table 5: **Per-task Success Rate (↑) of RoboSaGA** and the standard error of the mean with and without Saliency Buffer under visual domain shifts (BC-MLP).

| | *Buffer* | *w/o Buffer* |
|---|---|---|
| *In-domain* | $0.88_{\pm 0.02}$ | $\mathbf{0.88}_{\pm 0.02}$ |
| *Background* | $\mathbf{0.72}_{\pm 0.02}$ | $0.71_{\pm 0.02}$ |
| *Distractor* | $\mathbf{0.78}_{\pm 0.02}$ | $0.77_{\pm 0.02}$ |
| *mean* | $\mathbf{0.75}_{\pm 0.01}$ | $0.74_{\pm 0.01}$ |

Table 6: **Average Task Success Rate (↑) of RoboSaGA** with and without Saliency Buffer under VDS (BC-MLP).

| Buffer | No Buffer |
|---|---|
| 0.22s | 0.70s |

Table 7: Average Processing Time for Per-batch Augmentation

Here, we compare the performance and computational differences between utilising a saliency buffer and directly computing the saliency. In the RoboSaGA experiments, detailed in Sec. 4.1, 10% of the current batch is sampled for saliency updates and saved in the saliency buffer; the augmentation ratio $\alpha$ is maintained at 50% of the current batch. By contrast, when the buffer is disabled, saliency maps are directly computed for 50% of the batch. The experiments are conducted on the *Lift*, *Can*, and *Square* tasks in simulation using the BC-MLP policy.

As shown in *Tab.* 5, 6, the average success rates over three tasks for RoboSaGA, with and without the saliency buffer under visual domain shifts, are 0.75 and 0.74, respectively. No significant performance difference is observed. While the use of the saliency buffer does not enhance performance, it significantly reduces the computation time required for saliency extraction to one-third (*Tab.* 7).

# Appendix B: Experimental Details

## Task Descriptions

| Task | Action Dimension | Observation Dimension | Proprio. Dimension | Max. Steps | Long Horizon | Num. of Subtasks |
|------|------------------|----------------------|--------------------|-----------|--------------|------------------|
| | | Simulation | | | | |
| *Lift* | 7 | $2 \times 84 \times 84$ | 7 | 200 | No | 1 |
| *Can* | 7 | $2 \times 84 \times 84$ | 7 | 200 | No | 2 |
| *Square* | 7 | $2 \times 84 \times 84$ | 7 | 200 | No | 2 |
| *Transport* | 14 | $4 \times 84 \times 84$ | 7 | 200 | Yes | 8 |
| | | Real-world | | | | |
| *Toy* | 7 | $2 \times 84 \times 84$ | 12 | 160 | No | 2 |

Table 8: **Task Summary.** *Action Dimension*: Robot action dimension, including end-effector 6-DOF velocity and parallel gripper width. *Observation Dimension*: Number of views $\times$ Image observation dimension. *Proprio. Dimension*: the dimension of robot proprioceptive states. *Num. of Demos*: Number of demonstrations provided. *Maximum Steps*: Maximum rollout steps in simulation. *Long-Horizon*: Indicates whether the task requires learning multiple behaviours together. *Num. of Subtasks*: Number of sub-tasks within each task.

In this work, we utilise four simulation tasks from RoboMimic [7] with provided proficient human (PH) demonstrations, and one real-world pick-and-place task collected via proficient human teleoperation. All tasks employ Franka Panda as the manipulator within a 7-dimensional action space, which includes the 6 degrees of freedom for end-effector pose and gripper width. In the simulation, the proprioceptive states include the end-effector pose (rotation is represented by quaternion) and gripper width. The real task utilises a 12-dimensional robot state represented by a flattened SE(3) matrix (last row omitted). Except for the *Transport* task, all tasks use two camera views: a second-person camera and a first-person camera. The *Transport* task features two camera views associated with each manipulator. Task descriptions and details are summarised below and in *Tab*. 8.

- *Lift*: Grasp and lift a red cube from the table.
- *Can*: Grasp a Coke Can from the left bin (sub-task 1), and place it into the corresponding target bin on its right (sub-task 2).
- *Square*: Grasp the square nut by the handle (sub-task 1) and insert it into a matching square peg (sub-task 2).
- *Transport*:
  - Left arm: Pick the handle of the source bin's lid (sub-task 1), place the lid in the empty space (sub-task 2), grasp the hammer within the source bin (sub-task 3), and deliver it to the workspace within the right arm's reach (sub-task 4).
  - Right arm: Pick up a red cube from the target bin (sub-task 5), place it into the trash bin (sub-task 6), take the hammer delivered by the left arm (sub-task 7), and place it into the target bin (sub-task 8).
- *Toy*: Pick up a green squashy toy (sub-task 1) and place it into a yellow cup (sub-task 2). The location of the toy and the cup varies within a $10 \times 10 \, \text{cm}^2$ area.

## Policy and Training Details

BC-MLP and BC-RNN utilise the default network configurations as specified in RoboMimic [7]. Specifically, each image observation is processed by a `ResNet18` [27] followed by a spatial-softmax layer [28]. All observation features are then concatenated with the robot's proprioceptive states, forming a single feature vector. The flattened states are fed into either an MLP or an RNN. The MLP employs ReLU activations, while the RNN consists of a 2-layer LSTM. The final layer's

| Hyperparameter | BC-MLP | BC-RNN |
|---|---|---|
| Learning Rate (LR) | $1 \times 10^{-4}$ | $1 \times 10^{-4}$ |
| Actor MLP Dimensions | [1024, 1024] | - |
| RNN Hidden Dimension | - | 1000 |
| RNN Sequence Length | - | 10 |
| GMM Number of Modes | 5 | 5 |
| Image Encoder | ResNet18 | ResNet18 |
| SpatialSoftmax (num-KP) | 64 | 64 |

Table 9: Hyper-parameters for BC-MLP and BC-RNN.

hidden states are fed into the downstream Gaussian Mixture Model (GMM). As described in RoboMimic [7], during the rollout with GMM policies, the learned standard deviations of each mode are replaced with $1 \times 10^{-4}$. Hyper-parameter settings are detailed in *Tab.* 9.

*Diffusion Policy* employs the hybrid-CNN architecture as detailed in [2]. It utilises a similar image encoder as described above but replaces BatchNorm layers [29] with GroupNorm [30]. The input horizon, action horizon, and action prediction horizon are set to 2, 8, and 16, respectively. The learning rate is set to $1 \times 10^{-4}$.

Networks are defaulted to train from scratch with Adam [31] optimiser (learning rate set to $1 \times 10^{-4}$). Except for the simple *Lift* task is trained for 200 epochs, all tasks are trained for 600 epochs.

**Out-of-domain Images for RoboSaGA**

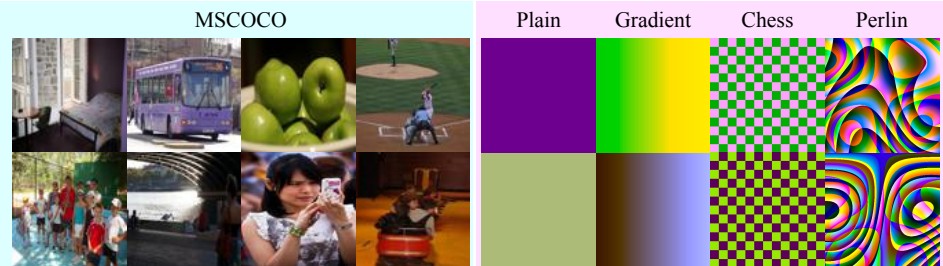

Figure 6: Examples of out-of-domain images for data augmentation

RoboSaGA utilises approximately 6,000 out-of-domain images for augmentation, comprising 5,000 real-world images from MSCOCO [26] and 1,000 synthetic images featuring plain, gradient, grid, chess, and Perlin patterns (*Fig.* 6). All images are subject to random rotations and brightness adjustments.

**Lighting and Shadow Evaluation Setup**

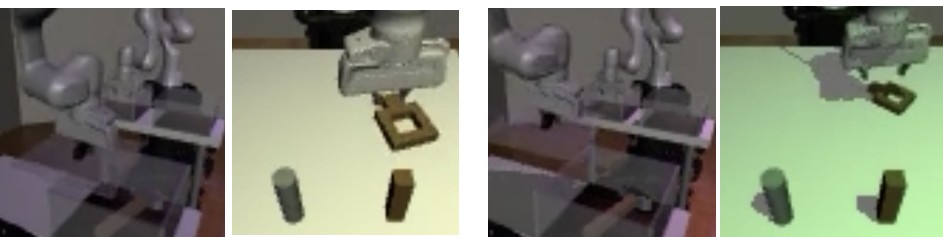

Lighting Changes + Shadow Rendering Off        Lighting Changes + Shadow Rendering On

Figure 7: **Examples of lighting and shadow variations in simulation evaluation.**

We conducted two sets of simulated experiments on various policies, including BC-MLP, BC-RNN, and diffusion policies, across the four simulated tasks, examining the impact of lighting changes (varying intensity and colour by adjusting the diffuse parameter between 0.2 and 0.8) with shadow rendering both enabled and disabled (shadows turned off in the training data). The experimental setup, tasks, and evaluation protocol are identical to those in the submitted version, with random cropping applied by default.

**Backgrounds and Distractors for Evaluation**

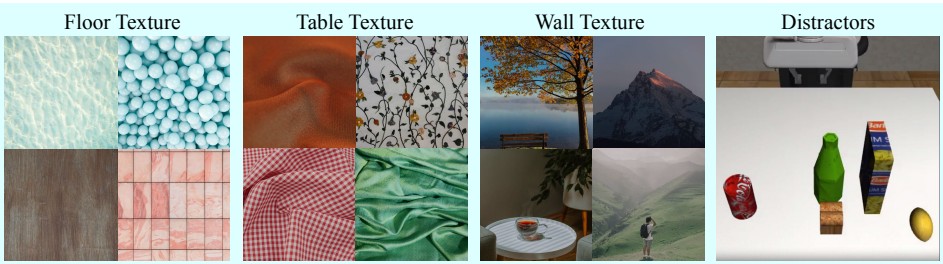

Figure 8: **Examples of Lighting and distractors in simulation evaluation.**

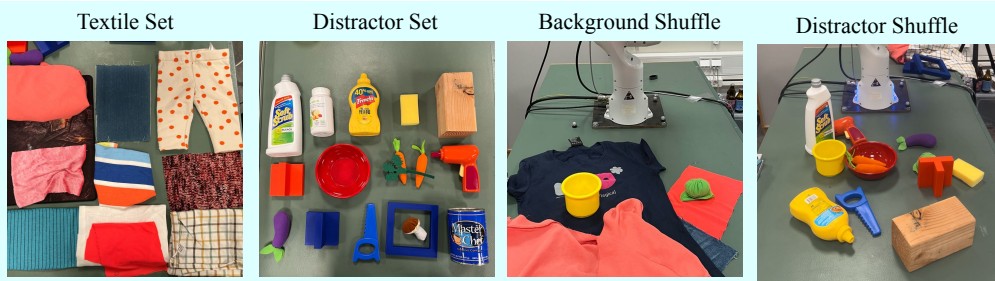

Figure 9: **Examples of textiles and distractors in real-world evaluation.**

In the *simulation*, as illustrated in *Fig*. 8, table textures are derived from images of textiles and patterns. Floor textures utilise common materials such as tiles and wooden flooring, while wall textures are selected from both indoor and outdoor scenes. Distractors, including items like a Coke can, bottle, cereal, bread, and lemon, form the default set. Some distractors may be removed to prevent duplicating task-specific objects. For example, the Coke can is excluded from the *Can* task. Distractors are strategically placed to minimise collisions with the manipulator and targets, although collisions are still possible. Each selected distractor has a 50% chance of appearing.

In the **real-world** setting, as shown in *Fig*. 9, background shuffling is achieved by varying the combinations of textiles drawn from the default textile set. These textiles sufficiently cover the field of view for both the second-person camera and the eye-in-hand camera, with examples of the respective fields of view illustrated in *Fig*. 4. Distractors are also drawn from the default distractor set, forming cluttered arrangements.

# Appendix C: Full Tables

All data points in the tables are presented as mean $\pm$ the standard error of the mean.

## 6.1 Simulated Experiments

| | | Crop | +Jitter | +SODA | +Over. | +SaGA |
|---|---|---|---|---|---|---|
| | In-domain | $0.97_{\pm0.01}$ | $\mathbf{1.00}_{\pm0.00}$ | $1.00_{\pm0.00}$ | $0.99_{\pm0.01}$ | $0.98_{\pm0.01}$ |
| Lift | Lighting | $0.27_{\pm0.04}$ | $0.97_{\pm0.01}$ | $0.97_{\pm0.01}$ | $\mathbf{0.99}_{\pm0.01}$ | $0.99_{\pm0.01}$ |
| | Lighting & Shadow | $0.21_{\pm0.03}$ | $0.75_{\pm0.04}$ | $0.94_{\pm0.02}$ | $\mathbf{0.97}_{\pm0.01}$ | $0.97_{\pm0.01}$ |
| | Background | $0.00_{\pm0.00}$ | $0.05_{\pm0.02}$ | $0.80_{\pm0.03}$ | $0.87_{\pm0.03}$ | $\mathbf{0.89}_{\pm0.03}$ |
| | Distractor | $0.21_{\pm0.03}$ | $0.31_{\pm0.04}$ | $\mathbf{0.96}_{\pm0.02}$ | $0.66_{\pm0.04}$ | $0.95_{\pm0.02}$ |
| | In-domain | $0.95_{\pm0.02}$ | $0.91_{\pm0.02}$ | $0.93_{\pm0.02}$ | $0.92_{\pm0.02}$ | $\mathbf{0.97}_{\pm0.01}$ |
| Can | Lighting | $0.42_{\pm0.04}$ | $0.69_{\pm0.04}$ | $0.93_{\pm0.02}$ | $0.93_{\pm0.02}$ | $\mathbf{0.98}_{\pm0.01}$ |
| | Lighting & Shadow | $0.19_{\pm0.03}$ | $0.53_{\pm0.04}$ | $0.93_{\pm0.02}$ | $0.87_{\pm0.03}$ | $\mathbf{0.95}_{\pm0.02}$ |
| | Background | $0.03_{\pm0.01}$ | $0.38_{\pm0.04}$ | $0.65_{\pm0.04}$ | $0.79_{\pm0.03}$ | $\mathbf{0.87}_{\pm0.03}$ |
| | Distractor | $0.43_{\pm0.04}$ | $0.33_{\pm0.04}$ | $0.57_{\pm0.04}$ | $0.53_{\pm0.04}$ | $\mathbf{0.71}_{\pm0.04}$ |
| | In-domain | $0.62_{\pm0.04}$ | $0.55_{\pm0.04}$ | $0.47_{\pm0.04}$ | $0.51_{\pm0.04}$ | $\mathbf{0.68}_{\pm0.04}$ |
| Square | Lighting | $0.02_{\pm0.01}$ | $0.43_{\pm0.04}$ | $0.59_{\pm0.04}$ | $0.59_{\pm0.04}$ | $\mathbf{0.62}_{\pm0.04}$ |
| | Lighting & Shadow | $0.00_{\pm0.00}$ | $0.09_{\pm0.02}$ | $0.46_{\pm0.04}$ | $0.33_{\pm0.04}$ | $\mathbf{0.51}_{\pm0.04}$ |
| | Background | $0.00_{\pm0.00}$ | $0.01_{\pm0.01}$ | $0.07_{\pm0.02}$ | $0.25_{\pm0.04}$ | $\mathbf{0.39}_{\pm0.04}$ |
| | Distractor | $0.37_{\pm0.04}$ | $0.37_{\pm0.04}$ | $0.41_{\pm0.04}$ | $0.42_{\pm0.04}$ | $\mathbf{0.69}_{\pm0.04}$ |
| | In-domain | $\mathbf{0.49}_{\pm0.04}$ | $0.47_{\pm0.04}$ | $0.41_{\pm0.04}$ | $0.37_{\pm0.04}$ | $0.46_{\pm0.04}$ |
| Transport | Lighting | $0.07_{\pm0.02}$ | $0.33_{\pm0.04}$ | $0.29_{\pm0.04}$ | $0.36_{\pm0.04}$ | $\mathbf{0.49}_{\pm0.04}$ |
| | Lighting & Shadow | $0.00_{\pm0.00}$ | $0.07_{\pm0.02}$ | $0.25_{\pm0.04}$ | $\mathbf{0.35}_{\pm0.04}$ | $0.33_{\pm0.04}$ |
| | Background | $0.00_{\pm0.00}$ | $0.00_{\pm0.00}$ | $0.01_{\pm0.01}$ | $\mathbf{0.05}_{\pm0.02}$ | $0.05_{\pm0.02}$ |
| | Distractor | $0.21_{\pm0.03}$ | $0.17_{\pm0.03}$ | $0.23_{\pm0.03}$ | $0.21_{\pm0.03}$ | $\mathbf{0.34}_{\pm0.04}$ |

Table 10: **Success Rate (↑) of BC-MLP** with Random Crop, Colour Jitter, Random Overlay, SODA and RoboSaGA under visual domain shifts.

| | | Crop | +Jitter | +SODA | +Over. | +SaGA |
|---|---|---|---|---|---|---|
| | In-domain | $0.97_{\pm0.01}$ | $0.97_{\pm0.01}$ | $0.95_{\pm0.02}$ | $\mathbf{1.00}_{\pm0.00}$ | $0.99_{\pm0.01}$ |
| Lift | Lighting | $0.43_{\pm0.04}$ | $0.93_{\pm0.02}$ | $0.95_{\pm0.02}$ | $\mathbf{1.00}_{\pm0.00}$ | $1.00_{\pm0.00}$ |
| | Lighting & Shadow | $0.17_{\pm0.03}$ | $0.66_{\pm0.04}$ | $0.95_{\pm0.02}$ | $\mathbf{0.99}_{\pm0.01}$ | $0.99_{\pm0.01}$ |
| | Background | $0.00_{\pm0.00}$ | $0.01_{\pm0.01}$ | $0.65_{\pm0.04}$ | $0.92_{\pm0.02}$ | $\mathbf{0.93}_{\pm0.02}$ |
| | Distractor | $0.21_{\pm0.03}$ | $0.19_{\pm0.03}$ | $0.92_{\pm0.02}$ | $0.96_{\pm0.02}$ | $\mathbf{1.00}_{\pm0.00}$ |
| | In-domain | $\mathbf{0.97}_{\pm0.01}$ | $0.97_{\pm0.01}$ | $0.96_{\pm0.02}$ | $0.97_{\pm0.01}$ | $0.97_{\pm0.01}$ |
| Can | Lighting | $0.65_{\pm0.04}$ | $0.85_{\pm0.03}$ | $0.94_{\pm0.02}$ | $\mathbf{0.95}_{\pm0.02}$ | $0.94_{\pm0.02}$ |
| | Lighting & Shadow | $0.40_{\pm0.04}$ | $0.81_{\pm0.03}$ | $\mathbf{0.93}_{\pm0.02}$ | $0.85_{\pm0.03}$ | $0.92_{\pm0.02}$ |
| | Background | $0.02_{\pm0.01}$ | $0.21_{\pm0.03}$ | $0.69_{\pm0.04}$ | $0.71_{\pm0.04}$ | $\mathbf{0.79}_{\pm0.03}$ |
| | Distractor | $0.32_{\pm0.04}$ | $0.39_{\pm0.04}$ | $0.64_{\pm0.04}$ | $0.55_{\pm0.04}$ | $\mathbf{0.70}_{\pm0.04}$ |
| | In-domain | $0.66_{\pm0.04}$ | $0.68_{\pm0.04}$ | $0.74_{\pm0.04}$ | $0.69_{\pm0.04}$ | $\mathbf{0.75}_{\pm0.04}$ |
| Square | Lighting | $0.03_{\pm0.01}$ | $0.66_{\pm0.04}$ | $0.68_{\pm0.04}$ | $0.73_{\pm0.04}$ | $\mathbf{0.82}_{\pm0.03}$ |
| | Lighting & Shadow | $0.00_{\pm0.00}$ | $0.25_{\pm0.04}$ | $0.64_{\pm0.04}$ | $0.62_{\pm0.04}$ | $\mathbf{0.73}_{\pm0.04}$ |
| | Background | $0.00_{\pm0.00}$ | $0.02_{\pm0.01}$ | $0.22_{\pm0.03}$ | $0.38_{\pm0.04}$ | $\mathbf{0.47}_{\pm0.04}$ |
| | Distractor | $0.33_{\pm0.04}$ | $0.40_{\pm0.04}$ | $0.38_{\pm0.04}$ | $0.59_{\pm0.04}$ | $\mathbf{0.72}_{\pm0.04}$ |
| | In-domain | $0.61_{\pm0.04}$ | $0.61_{\pm0.04}$ | $\mathbf{0.65}_{\pm0.04}$ | $0.59_{\pm0.04}$ | $0.57_{\pm0.04}$ |
| Transport | Lighting | $0.35_{\pm0.04}$ | $0.39_{\pm0.04}$ | $0.59_{\pm0.04}$ | $0.62_{\pm0.04}$ | $\mathbf{0.64}_{\pm0.04}$ |
| | Lighting & Shadow | $0.01_{\pm0.01}$ | $0.22_{\pm0.03}$ | $0.53_{\pm0.04}$ | $0.52_{\pm0.04}$ | $\mathbf{0.64}_{\pm0.04}$ |
| | Background | $0.00_{\pm0.00}$ | $0.00_{\pm0.00}$ | $0.05_{\pm0.02}$ | $0.14_{\pm0.03}$ | $\mathbf{0.20}_{\pm0.03}$ |
| | Distractor | $0.24_{\pm0.03}$ | $0.30_{\pm0.04}$ | $0.39_{\pm0.04}$ | $0.35_{\pm0.04}$ | $\mathbf{0.41}_{\pm0.04}$ |

Table 11: **Success Rate of BC-RNN** with Random Crop, Colour Jitter, Random Overlay, SODA and RoboSaGA under visual domain shifts.

| | Crop | +Jitter | +SODA | +Over. | +SaGA |
|---|---|---|---|---|---|
| **Lift** *In-domain* | $\mathbf{1.00}_{\pm 0.00}$ | $1.00_{\pm 0.00}$ | $1.00_{\pm 0.00}$ | $1.00_{\pm 0.00}$ | $1.00_{\pm 0.00}$ |
| *Lighting* | $\mathbf{1.00}_{\pm 0.00}$ | $1.00_{\pm 0.00}$ | $1.00_{\pm 0.00}$ | $1.00_{\pm 0.00}$ | $1.00_{\pm 0.00}$ |
| *Lighting & Shadow* | $\mathbf{1.00}_{\pm 0.00}$ | $1.00_{\pm 0.00}$ | $1.00_{\pm 0.00}$ | $1.00_{\pm 0.00}$ | $1.00_{\pm 0.00}$ |
| *Background* | $0.86_{\pm 0.03}$ | $0.86_{\pm 0.03}$ | $\mathbf{0.98}_{\pm 0.01}$ | $0.98_{\pm 0.01}$ | $0.93_{\pm 0.02}$ |
| *Distractor* | $0.55_{\pm 0.04}$ | $0.55_{\pm 0.04}$ | $\mathbf{1.00}_{\pm 0.00}$ | $1.00_{\pm 0.00}$ | $1.00_{\pm 0.00}$ |
| **Can** *In-domain* | $\mathbf{1.00}_{\pm 0.00}$ | $0.96_{\pm 0.02}$ | $0.97_{\pm 0.01}$ | $0.98_{\pm 0.01}$ | $0.97_{\pm 0.01}$ |
| *Lighting* | $0.93_{\pm 0.02}$ | $0.97_{\pm 0.01}$ | $0.93_{\pm 0.02}$ | $\mathbf{0.98}_{\pm 0.01}$ | $0.94_{\pm 0.02}$ |
| *Lighting & Shadow* | $0.80_{\pm 0.03}$ | $0.82_{\pm 0.03}$ | $0.93_{\pm 0.02}$ | $\mathbf{0.97}_{\pm 0.01}$ | $0.92_{\pm 0.02}$ |
| *Background* | $0.55_{\pm 0.04}$ | $0.72_{\pm 0.04}$ | $0.75_{\pm 0.04}$ | $0.87_{\pm 0.03}$ | $\mathbf{0.91}_{\pm 0.02}$ |
| *Distractor* | $0.56_{\pm 0.04}$ | $0.38_{\pm 0.04}$ | $0.77_{\pm 0.03}$ | $0.75_{\pm 0.04}$ | $\mathbf{0.86}_{\pm 0.03}$ |
| **Square** *In-domain* | $\mathbf{0.93}_{\pm 0.02}$ | $0.91_{\pm 0.02}$ | $0.87_{\pm 0.03}$ | $0.91_{\pm 0.02}$ | $0.91_{\pm 0.02}$ |
| *Lighting* | $0.45_{\pm 0.04}$ | $0.87_{\pm 0.03}$ | $0.85_{\pm 0.03}$ | $\mathbf{0.89}_{\pm 0.03}$ | $0.85_{\pm 0.03}$ |
| *Lighting & Shadow* | $0.12_{\pm 0.03}$ | $0.37_{\pm 0.04}$ | $\mathbf{0.83}_{\pm 0.03}$ | $0.78_{\pm 0.03}$ | $0.80_{\pm 0.03}$ |
| *Background* | $0.01_{\pm 0.01}$ | $0.15_{\pm 0.03}$ | $0.63_{\pm 0.04}$ | $0.59_{\pm 0.04}$ | $\mathbf{0.76}_{\pm 0.03}$ |
| *Distractor* | $0.48_{\pm 0.04}$ | $0.63_{\pm 0.04}$ | $0.81_{\pm 0.03}$ | $0.89_{\pm 0.03}$ | $\mathbf{0.90}_{\pm 0.02}$ |
| **Transport** *In-domain* | $0.91_{\pm 0.02}$ | $\mathbf{0.94}_{\pm 0.02}$ | $0.90_{\pm 0.02}$ | $0.90_{\pm 0.02}$ | $0.89_{\pm 0.03}$ |
| *Lighting* | $0.81_{\pm 0.03}$ | $\mathbf{0.87}_{\pm 0.03}$ | $0.80_{\pm 0.03}$ | $0.85_{\pm 0.03}$ | $0.81_{\pm 0.03}$ |
| *Lighting & Shadow* | $0.48_{\pm 0.04}$ | $0.63_{\pm 0.04}$ | $0.77_{\pm 0.03}$ | $\mathbf{0.85}_{\pm 0.03}$ | $0.82_{\pm 0.03}$ |
| *Background* | $0.00_{\pm 0.00}$ | $0.00_{\pm 0.00}$ | $0.33_{\pm 0.04}$ | $0.45_{\pm 0.04}$ | $\mathbf{0.58}_{\pm 0.04}$ |
| *Distractor* | $0.44_{\pm 0.04}$ | $0.52_{\pm 0.04}$ | $0.61_{\pm 0.04}$ | $0.58_{\pm 0.04}$ | $\mathbf{0.65}_{\pm 0.04}$ |

Table 12: **Success Rate (↑) of Diffusion Policy** with Random Crop, Colour Jitter, Random Overlay, SODA and RoboSaGA under visual domain shifts.

## 6.2   Real Experiments

| | BC-MLP | | | BC-RNN | | | Diffusion Policy | | |
|---|---|---|---|---|---|---|---|---|---|
| | *Crop* | *+Over.* | *+SaGA* | *Crop* | *+Over.* | *+SaGA* | *Crop* | *+Over.* | *+SaGA* |
| *In-domain* | $0.70_{\pm 0.11}$ | $\mathbf{0.85}_{\pm 0.08}$ | $0.85_{\pm 0.08}$ | $0.85_{\pm 0.08}$ | $\mathbf{0.95}_{\pm 0.05}$ | $0.90_{\pm 0.07}$ | $\mathbf{1.00}_{\pm 0.00}$ | $1.00_{\pm 0.00}$ | $1.00_{\pm 0.00}$ |
| *Background* | $0.00_{\pm 0.00}$ | $0.50_{\pm 0.11}$ | $\mathbf{0.75}_{\pm 0.10}$ | $0.00_{\pm 0.00}$ | $0.60_{\pm 0.11}$ | $\mathbf{0.85}_{\pm 0.08}$ | $0.00_{\pm 0.00}$ | $0.65_{\pm 0.11}$ | $\mathbf{0.75}_{\pm 0.10}$ |
| *Distractor* | $0.05_{\pm 0.05}$ | $0.65_{\pm 0.11}$ | $\mathbf{0.80}_{\pm 0.09}$ | $0.15_{\pm 0.08}$ | $0.65_{\pm 0.11}$ | $\mathbf{0.70}_{\pm 0.11}$ | $0.65_{\pm 0.11}$ | $\mathbf{0.95}_{\pm 0.05}$ | $0.95_{\pm 0.05}$ |

Table 13: **Real-world Success Rate (↑) for Each Policy**: Random Crop, Random Overlay, and RoboSaGA on the *Toy* task against distractor and background variations.

## Appendix D: Interpretability Misalignment in Saliency Maps

Here, we select three representative examples from the *Transport* task to illustrate the potential misalignment between input saliency from the policies and human interpretation.

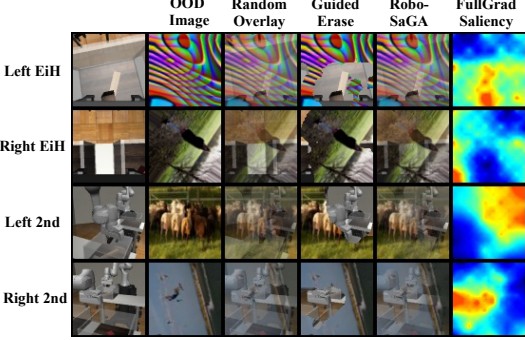

Figure 10: **Examples of Saliency Visualisations and Data Augmentations of *Transport* Task** from BC-MLP. *EiH*: eye-in-hand camera. *2nd*: second-person-camera

As observed in Fig. 10, the saliency maps produced by the history-independent BC-MLP align more closely with human intuition compared to those from BC-RNN (Fig. 11a) and the Diffusion

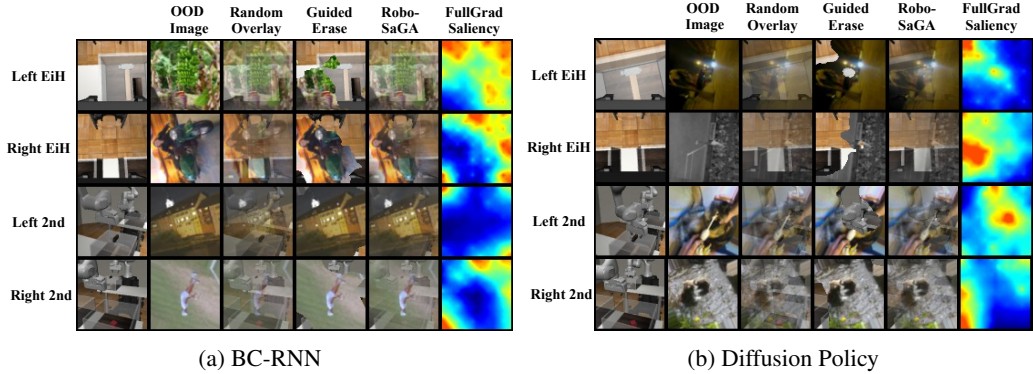

Figure 11: **Examples of Saliency Visualisations and Data Augmentations of *Transport* Task** from BC-RNN and Diffusion Policy. *EiH*: eye-in-hand camera. *2nd*: second-person-camera

Policy (Fig. 11b). In these visualisations, the Left Eye-in-Hand (EiH) camera focuses on the box handle, while the Right EiH camera does not provide task-critical information, exhibiting random focus instead. The left second-person camera focuses on the right arm, and the right second-person camera focuses on the right hand. Notably, the left second-person camera does not focus on the lid handle, as this piece of information is already provided by the left EiH camera. However, despite the alignment of the saliency maps with human intuition, experimental results indicate that this alignment does not necessarily translate into improved robustness against background variations. Indeed, none of the tested augmentation methods enhanced the performance of BC-MLP in the *Transport* task (see *Tab*. 10, 11, 12).

Although achieving higher robustness against background variations, the saliency maps produced by the history-dependent BC-RNN and Diffusion Policy, in contrast to the history-independent BC-MLP, are less interpretable from a human perspective. These maps can focus on elements considered trivial by humans. Given that the task relies on historical observations, the robot's states, multi-views, and exhibits varying levels of visual dependency throughout its execution, we argue that what humans perceive as visually important may not always align with the network's focus.

