# OpenReview forum: "Enhancing Visual Domain Robustness in Behaviour Cloning via Saliency-Guided Augmentation"
_robot-learning.org/CoRL/2024/Conference — CoRL 2024_

### Official Review · Reviewer_wTGw · 2024-07-08
**Application of well-studied saliency based augmentation for behavior cloning**

**Originality:** 3
**Technical Quality:** 3
**Clarity Of Presentation:** 4
**Potential Impact:** 3
**Recommendation:** 3
**Confidence:** 3

**Review:**

The bird's eye view of this paper is a strategy to apply overlay based augmentation techniques that do well in computer vision tasks to improve training performance for the task of behavior cloning. However, since you want to be careful and not augment parts of the image which are important, the main idea is to compute saliency maps over the image to identify the important regions, and then weigh the augmentation strategy based on this saliency map.

Overall, this paper is clear and well-explained. The authors apply a method similar to KeepAugment specifically for behavior cloning and show that it improves accuracy tremendously when there are visual domain shifts. The results and findings are quite interesting

Strengths:
1. The technical approach is well-explained with good explanations on the motivations behind each experiment.
2. There aren’t many papers on such augmentation techniques for behavior cloning, which might make this informative to the community as a whole
3. They also demonstrate their results on hardware and show that there are significant improvements

Weaknesses:
1. The method used is very similar to KeepAugment, which is already a well-known method in computer vision
2. Further analysis/references need to be provided for the other claim of the paper that non-overlay techniques are not effective. Between overlay techniques, the claims made in this paper hold but it's unclear how this augmentation would perform relative to other techniques.

**Quality Of The Limitations Section:**

3

**Questions For Rebuttal:**

A few major points:
1. A discussion on fullgrad in the related work section would help the reader understand the motivation behind using this over other saliency methods.
2. As per my understanding, consecutive images within a trajectory could get very different augmentations, i.e. different backgrounds which is unlikely to happen in real-world scenarios. Wouldn’t this mean that data could be out of distribution for consecutive images making saliency maps for images later in the trajectory a little arbitrary, especially for BC-RNN and Diffusion Policy?
3. It would be interesting to see how the saliency maps look for time dependent networks, especially BC-RNN and Diffusion Policy in the main paper as well
4. Does it make sense to use this method in the full training cycle? The saliency map might not make sense in the very beginning - would this cause slower convergence?
5. The abstract mentions that pixel shift is not effective for visual domain shifts, but would like to see this backed up by citations or experiments. All the experiments compare only overlay based methods with each other

More minor changes/typos:
1. Line 211 - "Another unspecified method" is a typo
2. Line 255 - "Distil" -> "Distill"
3. Line 166 - "Thus" should be "this"

**Robotics Focus:**

4

**Summary Of Paper:**

The authors' primary assertion is that traditional augmentations like pixel shift are effective in boosting in-domain performance; however, these methods are not as effective for visual domain shifts (i.e. shifts where there are other objects/background is different). To improve performance for visual domain shifts, it is crucial to superimpose additional objects or images onto the training image. However, this approach can result in information loss for behavior cloning. Therefore, the authors propose a saliency-guided approach to augmenting existing data.

**Summary Of Recommendation:**

The paper is very clear and easy to follow. The performance improvement is fairly significant and the experiments performed are thorough. Some additional analysis could be provided to improve the overall quality. In terms of originality though such methods have already been explored or published in the field of computer vision, but not in the context of behavior cloning. Hence my recommendation for a weak accept.

---

### Official Review · Reviewer_Nf11 · 2024-07-16
**A saliency-guided augmentation strategy for BC**

**Originality:** 3
**Technical Quality:** 3
**Clarity Of Presentation:** 4
**Potential Impact:** 3
**Recommendation:** 3
**Confidence:** 3

**Review:**

The idea of using saliency to guide superimposition-based image augmentation makes intuitive sense -- When introducing augmentation signals, the method aims to reduce the obscuring of the task-related information by avoiding excessive overlay in regions of interest. The saliency is extracted by the Fullgrad method. These augmentation methods are used to improve the performance/robustness of BC-based policy learning.

Experimental validation is conducted on several simulation-based robotic manipulation environments and a real-world robotic manipulation task. The proposed method generally shows better performance than an overlay baseline and another augmentation method -- SODA.

The proposed method is of interest to the community in achieving generally applicable manipulation policies in diverse visual environments. This reviewer believes the notion of task-related information could be quite extendable -- e.g. when having access to a very good segmentation model, the segmentation map could also be used for deciding where to do overlay or not. As the authors also noted, the saliency map is not always a good proxy for task relevance.

Overall, the paper is well-written and shows convincing/interesting results. The discussions on limitations are appreciated.

**Quality Of The Limitations Section:**

3

**Questions For Rebuttal:**

I think the paper in its current form may have a limited impact due to the potential problems with saliency quality. It would be interesting to extend the scope of the paper to more general task-relevance notions which this reviewer belives would greatly improve the applicability/significance of the paper.

For the simulated environment task, would it be possible to extract the ground-truth task-relevant pixels? This would be an interesting baseline to consider and somewhat establish an upper bound of task-relevance-guided overlay. In practice, such maps could potentially be obtained through image segmentation models in certain domains.

**Robotics Focus:**

4

**Summary Of Paper:**

The authors proposes to guide superimposition-based augmentation with model saliency with validation on simulation/real-world manipulation tasks

**Summary Of Recommendation:**

Recommending weak accept -- happy to raise my scores if there would be clarifications/experiments to the questions/comments above.

---

### Official Review · Reviewer_Kp3m · 2024-07-20
**Review of paper 464**

**Originality:** 3
**Technical Quality:** 3
**Clarity Of Presentation:** 5
**Potential Impact:** 3
**Recommendation:** 3
**Confidence:** 3

**Review:**

The authors executed a really cool idea and I want to start by commending them! The idea to use saliency maps to ``force'' the policy to attend to action relevant information is clever and they show meaningful improvements over SODA in the presence of new distractors and backgrounds.

However, to fully verify the experimental claims, the authors should address the following concerns.

1. There is not enough information about the data augmentations used in the baseline policy. The paper points out that the invariance assumptions of standard image augmentation techniques from computer vision do not hold (e.g., I can't horizontally flip an image), but there are many that do and these provide good robustness guarantees. I would expect the baseline policy to use something like RandomCrop, ColorJitter, and RandomRotation. [1, 2] show that these augmentations are important in getting good OOD performance.

2. Why was lighting excluded as a test-time distribution shift? Prior work studying the problem of OOD policy performance tests lighting and it causes a non-trivial drop in performance [1, 2]. Lighting is also the only shift that affects the manipulation relevant artifacts (e.g., the robot arm and the target object) so it would be especially interesting to see how well this augmentation works in lighting shifts.

Also, I'm not sure if the number of gradient steps between the baseline and RoBoSaGA are fair. RoBoSaGA _sort of_ gets extra gradient steps  in the computation of the saliency map. I'm curious to hear the other reviewers thoughts on this.

Finally, I think it's worth considering the computational expense of RoBoSaGA. The saliency maps are expensive to compute, which makes the asymptotic of there approach a little prohibitive.

Here are more suggestions for improvement. They did not factor into my recommendation:
- epsilon is weird notation in (2) because I'm used to seeing it refer to a small value, not an ood image.
- in line 62, does "visual modality" refer to camera viewpoint? This wording confused me.
- the first paragraph of the related work kind of feels like a preliminaries section. Lines 68-71 are also too broad for the paper. A better RW section would be a section on robustness of BC policies.
- It would be great to see the time complexity of this method vs baselines in a table somewhere, maybe in the appendix.
- I think it's cool that the saliency maps make the augmentation super policy relevant! I wish the authors discussed and analyzed this more.
- line 255: should distil be distill?

[1] Decomposing the generalization gap in imitation learning for visual robotic manipulation. Xie et. al. 2023. https://arxiv.org/pdf/2307.03659.
[2] What Makes Pre-Trained Visual Representations Successful for Robust Manipulation? Burns et. al. 2023. https://arxiv.org/abs/2312.12444.

**Quality Of The Limitations Section:**

3

**Questions For Rebuttal:**

1. What data augmentations are used in the baseline model and with what parameters? How well does a policy with RandomCrop, ColorJitter, and RandomRotation perform?

2. How does RoboSaGA perform under lighting distribution shifts?

**Robotics Focus:**

4

**Summary Of Paper:**

The author propose an image augmentation scheme called RoboSaGA for policy learning. The augmentation is $I = M \odot x + (1-M) \odot x_{\text{ood}}$ where the mask, $M$ comes from a saliency map. This augmentation decreases performance degradation in the presence of background changes and the presence of distractors. They test their augmentation with BC-MLP, BC-RNN, and Diffusion Policy across 3 Robomimic tasks and one real-world task.

**Summary Of Recommendation:**

Leaning reject unless more answers can be provided about lighting shifts and baseline data augmentations.

---

### Author Rebuttal · Authors · 2024-08-09

This zip file includes a single pdf, which contains all replies to reviewers, figures, and full tables.

---

### Decision · Program_Chairs · 2024-09-04

**Decision:**

Accept

**Comment:**

AC summarizes the strengths and weaknesses of the paper based on the reviewers' comments as follows:

**Strength:**

- The paper is overall well-written
- The idea of using saliency for data augmentation is reasonable
- The results reported in the paper are promising.

**Weakness:**

- The idea is similar to that of KeepAugment
- There are some unclear points regarding experiments

=== comments after the rebuttal ===

In the rebuttal, the authors provided additional experiments on lighting variations, which demonstrate the advantage of the proposed method. The reviewers appreciate the authors' response and agree that the paper meets the criteria of CoRL.